# Impact of RAMPA Therapy on Nasal Cavity Expansion and Paranasal Drainage: Fluid Mechanics Analysis, CAE Simulation, and a Case Study

**DOI:** 10.3390/biomimetics11010005

**Published:** 2025-12-23

**Authors:** Mohammad Moshfeghi, Yasushi Mitani, Yuko Okai-Kojima, Bumkyoo Choi

**Affiliations:** 1Department of Mechanical Engineering, Sogang University, Seoul 04107, Republic of Korea; 2Codomo Clinic, Tokyo 180-0004, Japan; mitani@trust.ocn.ne.jp; 3Children and Women Dental Clinic, Tokyo 106-0046, Japan

**Keywords:** RAMPA, paranasal sinuses, mucus drainage FEM, CFD, empyema

## Abstract

**Background:** Impaired mucus drainage from the paranasal sinuses is often associated with nasal obstruction and reduced airway function in growing patients. Orthopedic maxillary protraction and expansion techniques can enhance airway dynamics, but their underlying fluid–structure mechanisms remain insufficiently understood. **Objective:** To validate that the Right Angle Maxillary Protraction Appliance (RAMPA), combined with a semi-rapid maxillary expansion (sRME) intraoral device gHu-1, improves mucus drainage by enhancing nasal airflow through nasal cavity expansion. **Methods:** The effects of RAMPA therapy were analyzed using computational fluid dynamics (CFD) for single-phase (air) and two-phase (air–mucus) flows within the nasal cavity, employing the unsteady RANS turbulence model. Finite element method (FEM) results from prior studies were synthesized to assess changes in the center and radius of maxillary rotation induced by RAMPA-assisted sRME. A male patient (aged 8 years 7 months to 11 years 7 months) treated with extraoral RAMPA and the intraoral appliance (gHu-1) underwent pre- and post-treatment cone-beam computed tomography (CBCT) and ear, nose, and throat (ENT) evaluation. **Results:** FEM analysis revealed an increased radius and elevated center of maxillary rotation, producing expansion that was more parallel to the palatal plane. CFD simulations showed that nasal cavity expansion increased airflow velocity and pressure drop, enhancing the suction effect that promotes mucus clearance from the frontal sinus. Clinically, nasal passages widened, paranasal opacities resolved, and occlusal and intermolar widths improved. **Conclusions:** RAMPA combined with sRME improves nasal airflow and maxillary skeletal expansion, facilitating paranasal mucus clearance and offering a promising adjunctive approach for enhancing upper airway function in growing patients.

## 1. Introduction

Maxillary expansion devices have been developed to treat maxillary hypoplasia. As early as the 1950s, Derichsweiler [1] and Gerlach [2] renewed interest in rapid maxillary expansion (RME) by reporting improvements in nasal respiration and increases in the width of the maxillary apical base following expansion. Slow maxillary expansion (SME) was later introduced as a treatment aimed at reducing the side effects of RME by gradually expanding the maxilla with lower forces. In 1977, Mew introduced semi-rapid maxillary expansion (sRME) as a remedy for the disadvantages of both RME and SME therapies [3]. Over the years, sRME has demonstrated its effectiveness in patients of various ages. For example, Işeri applied sRME in patients ranging from a 3-year-old child to adolescents and showed that both dental and skeletal changes remained stable [4].

Because maxillary expansion can influence nasal structures, the study of airway dimensions has long been an area of interest for orthotists and an important diagnostic consideration in several dental subspecialties. This attention arises from the potential impact of increased airway resistance on abnormal growth patterns and the heightened risk of conditions such as empyema in children [5,6,7]. For example, research has shown that the nasal cavity may widen by 1.9 mm overall, but can expand as much as 8–10 mm at the level of the inferior turbinates, while more superior areas may move medially during RME [8]. However, medial movement in the superior region presents a problem because of the presence of the ethmoid and sphenoid sinuses in that area.

To better understand these structural and functional effects, researchers over the past two decades have conducted finite element method (FEM) studies on s-RME using a relatively new appliance called the Right Angle Maxillary Protraction Appliance (RAMPA), shown in Figure 1. Owing to its unique geometry, RAMPA has been demonstrated to exert an anterosuperior protraction force on the maxilla, enabling correction of occlusal discrepancies while helping to counteract gravitational influences. RAMPA has also shown promise due to its modular design and its ability to interface with various intraoral devices, allowing simultaneous s-RME treatment and maxillary protraction [9,10,11,12,13]. This combined functionality contributes to the expansion of the nasal passages. As the nasal passages widen, their resistance decreases, resulting in increased airflow velocity. As shown later in this section, this implies that following treatment with RAMPA, the same breathing effort produces a larger volumetric flow rate and a higher average airspeed through the expanded passage.

Understanding how airflow behaves in these expanded passages is essential because mucus produced in the paranasal sinuses—including the maxillary (MS), anterior ethmoid (AE), posterior ethmoid (PE), and sphenoid sinuses (SpS)—drains into the nasal cavity through their respective ostia. A schematic view and detailed explanations of these pathways are provided in Figure 2 and Table 1.

From a fluid mechanics perspective, if the airflow inside the nasal cavity is idealized as inviscid (frictionless) and incompressible, Bernoulli’s equation (Equation (1)) states that energy is conserved along a streamline.(1)$$P+\frac{1}{2}\rho v^{2}+\rho gz=constant$$

Since the height difference z is on the order of only a few millimeters, its impact on the Bernoulli equation is negligible. Thus, Bernoulli’s equation simplifies to state that, along a streamline, the pressure decreases as the flow speed increases (Equation (2)).(2)$$P+\frac{1}{2}\rho v^{2}=constant$$

Figure 3 illustrates a real-world application of Bernoulli’s equation using a nebulizer. As air passes through the horizontal pipe of the nebulizer, its velocity increases, causing the pressure to fall below atmospheric pressure. Consequently, vapor particles from the container are drawn into the horizontal pipe and carried toward the patient’s nose.

Now, if one instead assumes that the flow is viscous (while still Newtonian, incompressible, and laminar), the Hagen–Poiseuille equation for a long, straight, circular tube provides the fundamental relationship between pressure drop and volumetric flow rate:(3)$$Q=\frac{\pi r^{4}}{8\mu L}∆P$$

The corresponding average velocity is as follows:(4)$$V_{avg}=\frac{Q}{\pi r^{2}}=\frac{r^{2}}{8\mu L}∆P$$
indicating that, for a fixed driving pressure ∆P (i.e., similar breath effort), $V_{\mathrm{avg}}\;$scales with $r^{2}$. Thus, even modest increases in the effective radius of the nasal passage can produce disproportionately large increases in both volumetric flow rate and average airspeed. Returning to the Bernoulli equation (Equation (2)), a higher flow velocity is associated with a reduction in local static pressure; therefore, an expanded nasal cavity not only permits greater airflow for the same effort but also enhances local suction effects within the passages.

Even if the flow inside the nasal passages is assumed to be turbulent, the Darcy–Weisbach equation (Equation (5)), which provides a general formulation for pressure loss due to friction in viscous flow, leads to Equation (6), indicating that $V_{\mathrm{avg}}$ increases approximately with $\sqrt{r}$. Thus, for a fixed driving pressure (∆P), a larger effective diameter still tends to yield a higher average velocity.(5)$$∆P=f\;\frac{L}{r}\;\frac{\rho V_{avg}^{2}}{4}$$(6)$$V_{avg}=\sqrt{\frac{4r∆P}{\rho fL}}$$

As shown in the above discussion, regardless of which flow assumption is applied to airflow analysis within the nasal cavity, the same overall phenomenon emerges. The anterosuperior maxillary expansion produced by RAMPA increases the dimensions of the nasal passages, thereby reducing airflow resistance and leading to higher airspeed within the cavity. Consequently, after treatment with RAMPA, the same breath effort generates a larger volumetric flow rate and a higher average velocity through the widened passages. The fluid–mechanical basis for this change is summarized below.

Beyond the positive effects of increased airflow velocity on reducing airway resistance, higher intranasal airspeed also improves mucus drainage due to the shear-thinning properties of mucus. Research demonstrates that mucus is a non-Newtonian fluid and exhibits log-linear shear-thinning behavior, meaning it is highly viscous at low shear rates but becomes considerably less viscous as shear rates increase. Under physiological shear rates (e.g., $10^{3}$–$10^{4}\;s^{-1}$), mucus viscosity decreases dramatically, allowing it to flow more easily. This shear-thinning behavior has been reported by Jory et al. [14]. Tauwald et al. [15] likewise confirmed that mucus behaves as a non-Newtonian, shear-thinning, non-linear viscoelastic fluid—an expected characteristic of complex biological gels. In addition, Rajendran et al. [16] conducted computational and experimental studies of mucus clearance and explicitly modeled mucus as a shear-thinning (power-law) fluid, further supporting its non-Newtonian viscosity reduction under shear. Several other studies [17,18] have emphasized that shear-thinning behavior is a critical feature of the airway mucociliary system, helping reduce mucus viscosity and facilitating its clearance.

In addition to rheological effects, another important aspect of nasal airflow is its cyclic nature during inhalation and exhalation. Computational and experimental studies consistently indicate that the dynamic airflow generated during normal breathing promotes ventilation and exchange within the paranasal sinuses, despite their connection to the nasal passage through narrow ostia. CFD simulations performed by Tretiakow et al. [19] on human sinonasal geometries demonstrated that each respiratory cycle produces bidirectional and multidirectional flow patterns through the maxillary sinus ostium, enabling fresh air exchange and gas renewal within a single breath. More recent pulsatile-flow simulations by Warfield-McAlpine et al. [20] revealed that sinusoidal and pulsating airflow enhances reverse-flow components and shear stresses in the ostiomeatal complex—mechanisms capable of thinning mucus and promoting its movement from the sinus cavity into the nasal passage.

The physiological relevance of these oscillatory flows is further supported by studies coupling CFD with particle tracking and nasal wall compliance. Research by Modaresi et al. [21] and Jiao et al. [22] suggests that airflow-induced shear, particularly in the middle meatus and maxillary ostium, can augment mucociliary clearance by reducing mucus viscosity and maintaining airway hydration. Although ciliary motion remains the primary driver of mucociliary transport, small-amplitude oscillatory airflow contributes to secondary clearance pathways and helps prevent mucus stasis [23]. When combined with the shear-thinning behavior of nasal mucus, these oscillations may therefore act synergistically to facilitate drainage and maintain sinus health. Collectively, these findings support the view that the cyclic pressure and velocity oscillations associated with respiration create favorable hydrodynamic conditions for mucus transport.

Having addressed the theoretical fluid mechanics foundations underlying the positive influence of RAMPA on mucus drainage, the remainder of this study focuses on quantitative evidence of these effects. This includes finite element method analysis of maxillary expansion, computational fluid dynamics simulations of airflow and mucus behavior, and a clinical study evaluating the functional outcomes of RAMPA treatment.

## 2. Materials and Methods

This section presents the FEM simulation details for the RAMPA appliance, followed by a description of the CFD modeling framework, and concludes with the clinical treatment protocol and associated data.

### 2.1. Finite Element Method

In the FEM in the present study, a removable intraoral appliance (gHu-1) was simulated in combination with the RAMPA extraoral component. Static linear FEM was employed as the computer-aided engineering tool. A 3D model of a skull replica (3B Scientific Co., GmbH, Hamburg, Germany), consisting of 22 bones, all craniofacial sutures, teeth, and periodontal ligaments (PDLs), served as the basis for the analysis. The skull model was scaled to match the dimensions of the patient in the accompanying clinical case. The FEM closely followed the modeling framework used in previous studies by the present authors [9,10,11,12,13]. Owing to craniofacial symmetry, only one half of the skull was simulated.

The finite element model included cortical and cancellous bone structures as well as the midpalatal suture (MPS). Bone thicknesses were assigned according to values reported in prior investigations [24,25,26,27]. All sutures, including the MPS, were meshed using at least five elements across their thickness, ensuring adequate flexibility in all deformation modes—an essential requirement from a structural mechanics standpoint. Material properties for bones and sutures were assumed to be isotropic and were adopted from previous research, as summarized in Table 2 [28,29].

The FEM mesh consisted of approximately 1.1 million tetrahedral elements and 1.8 million nodes, with finer meshes applied to the sutures. Figure 4 shows the mesh arrangement used for the present FEA (for improved visualization, the mesh resolution shown in these images has been reduced by half).

Returning to Figure 1a–c, during RAMPA therapy, six nodal forces are applied to the device. These include two horizontal (anterior) forces on the front hooks and four vertical (superior) forces—two on the front hooks and two on the side hooks—all applied via elastic rubber bands. The magnitudes of these forces, determined through a series of tensile tests, were selected based on the same research team’s previous studies [9]. The force values and RAMPA dimensions ($L_{1}$ to $L_{6}$) are listed in Table 3.

Given the FEM model’s bilateral symmetry, a symmetric boundary condition was applied along the midline of the MPS, constraining that plane to displace only within the sagittal direction. In addition, the posterior region of the foramen magnum and the coronal suture were fixed with zero displacement in the x, y, and z directions. Boundary conditions of the FEM simulation are shown in Figure 5.

### 2.2. Computational Fluid Dynamics Simulations

A 3D CFD model of the airway and paranasal sinuses was generated from a CT-derived geometry (Figure 6). As shown in Figure 7, the model included the nasal cavity, nasopharynx, and the maxillary, frontal, ethmoid (anterior and posterior), and sphenoid sinuses. The CFD domain extends from the nostrils to a region near the throat (pharynx).

It should be noted that although the model follows the natural anatomy of the nasal passages, it was intentionally modified to include localized bends and narrow passages, as shown in Figure 7. These adjustments were introduced primarily to support the mucus-drainage simulations, based on the understanding that mucus viscosity exerts a more pronounced influence in smaller cross-sectional areas. To avoid constructing separate CFD models, and to minimize variability arising from differences in mesh configuration, all geometric variations were incorporated into a single CFD model.

The geometry was meshed with 790,000 polyhedral cells, with a maximum cell size of 0.7 mm and prism layers with a resolution of 0.06 mm and an expansion ratio of 1.2.

The boundary conditions applied to the CFD model consisted of a uniform flow velocity at the throat and a free inlet with atmospheric pressure at the nostrils, ensuring satisfaction of mass conservation.

A Volume of Fluid (VOF) method was employed for the air-mucus simulations. Air and mucus densities and dynamic viscosities were set to 1.184 kg/m^3^ and 1.84 × 10^−5^ Pa·s for air, and 1000 kg/m^3^ and 10 Pa·s for mucus, respectively [30]. For both the air-only and VOF simulations, unsteady RANS simulations with a k–ε turbulence model were used. The simulations were performed using a fixed time step of 0.001 s, with 10 inner iterations per step applied.

### 2.3. Clinical Study

The clinical study is presented as a case report describing the outcomes of RAMPA therapy used in combination with the gHu-1 appliance for the treatment of a boy. The patient underwent RAMPA therapy from age 8 years 7 months to 11 years 7 months. The resulting clinical data were used to compare the predicted mechanisms with the observed treatment-induced improvements.

The expansion screw followed the semi-rapid maxillary expansion protocol described by Mew [3], which allows gradual orthopedic adaptation and reduces the risk of relapse. The screw was activated by 1/8 turn every 2–3 days, producing approximately 1.5 mm of expansion per week.

An experienced orthodontist (TS) conducted the evaluation by generating two independent tracings from the cephalometric radiographs and subsequently performing the cephalometric analysis. Cone-beam computed tomography (CBCT) was used to evaluate dimensional changes in the upper airway, skeletal facial structures, and hyoid bone position.

## 3. Results and Discussions

### 3.1. FEM Analyses of RAMPA’s Impact on Nasal Cavity Expansion

Building on the previous discussions of RAMPA mechanics, this section draws on the authors’ earlier FEM studies to further substantiate those findings. Accordingly, the lateral expansion of the maxilla is examined using points A–E (Figure 8).

The external forces applied to the RAMPA (Figure 1) result in an increased radius of rotation and a shift of the effective center upward toward the frontozygomatic region. As shown in Figure 9a–c, this mechanical effect produces greater lateral expansion at all measured points.

While the above discussion provides a simplified interpretation of the mechanics, it is supported by the team’s 3D FEM results using a sutured craniofacial model. Table 4 lists midpalatal suture displacements for a gHu-1 simulation with RAMPA [11]. As observed in Table 4, (1) RAMPA induces lateral expansion of the craniofacial bones, and (2) produces nearly uniform, parallel displacements along the palatal plane. When these structural findings are correlated with the two-phase flow simulation results, it becomes evident that RAMPA therapy not only widens the nasal passages but also maintains a smooth, continuous passage geometry without abrupt curvature or diameter changes. Both effects act synergistically to enhance mucus-drainage efficiency.

### 3.2. CFD Simulations of Air and Air–Mucus Transport

This section is divided into three subsections. The first examines how RAMPA-induced anatomical changes affect airflow speed inside the nasal cavity. The second presents two-phase Volume of Fluid (VOF) simulations representing the transport of an air–mucus mixture and the drainage of mucus from the paranasal sinuses. The third subsection focuses on airflow driven by a sinusoidal inhalation–exhalation profile, highlighting the effects of respiration on flow patterns and pressure fluctuations within the nasal cavity.

#### 3.2.1. Effect of Air Speed on Local Pressure Inside the Nasal Cavity

The investigation in this subsection focuses on how increases in airflow velocity influence local pressure within the nasal cavity. As a foundational step, a constant throat flow speed of $V_{\mathrm{th}}=1\;m/s$ was simulated, with air entering freely through the nostrils to satisfy mass conservation. This produced a volumetric flow rate of approximately 4 L/min, consistent with the findings of Fleming et al. [31]. Additional simulations were performed at throat velocities of $V_{\mathrm{th}}=1.2\;m/s$ and 1.5 m/s. These simulations were used to evaluate pressure and velocity distributions throughout the domain, including sample points P and Q as shown in Figure 7.

Figure 10 compares the resulting velocity and pressure contours, using identical color scales for each parameter. As shown, airflow velocity inside the nasal passage exceeds that in the throat (laryngeal) region, which reflects the larger cross-sectional area at the throat. Furthermore, comparing the vertical passage connected to the frontal sinus (located anteriorly) with the passage descending from the ethmoidal and sphenoidal sinuses (located posteriorly) reveals notable spatial variations in pressure drop from the nostrils toward the larynx. This results in a substantial pressure difference between points P and Q.

From the above figures, it is clear that pressure values decrease from the nostrils toward point P, where the velocity increases. The same behavior is observed at point Q. The corresponding values are listed in Table 5. This pressure drop generates a local suction effect at these points, which contributes to improved mucus drainage from the frontal sinus.

Evaluating the above values shows that, in all three cases, the ratio of the air speed at point P to the air speed at the throat (or nostril) remains constant, which is consistent with the conservation of mass. However, substituting these values into Bernoulli’s equation reveals that the equation is not satisfied perfectly. This discrepancy arises because the CFD simulations incorporate wall friction and allow turbulent flow, resulting in pressure drops that differ from those predicted by the idealized Bernoulli equation.

Furthermore, Figure 11 presents the velocity and pressure contours for each simulation, displayed using their respective local minima and maxima. As shown, when the throat velocity is $V_{\mathrm{th}}=1.0\;m/s$, the maximum airspeed inside the cavity reaches 5.6 m/s. For $V_{\mathrm{th}}=1.2\;m/s$ and 1.5 m/s, the maximum speeds increase to 6.7 m/s and 8.1 m/s, respectively. A similar trend is observed in the pressure contours: the pressure difference between the nostril and the throat increases from 162.4 Pa (for $V_{\mathrm{th}}=1.0\;m/s$) to 201 Pa (for $V_{\mathrm{th}}=1.2\;m/s$), and finally to 257 Pa (for $V_{\mathrm{th}}=1.5\;m/s$). These values underpin the conclusion that nasal cavity expansion substantially influences airflow within the nasal passages and, consequently, sinus drainage.

#### 3.2.2. Effect of Sinusoidal Respiration on Flow Fluctuation Inside Nasal Cavity

In this section, a sinusoidal breathing rate of 20 breaths per minute (BPM) [31] is simulated by applying the following velocity profile at the throat to mimic inhalation and exhalation.(7)$$V\left(t\right)=V_{th}\sin\left(2\pi ft\right)$$
where $V_{th}=$ 1.0, 1.2 and 1.5 m/s, and *f* = 1/3 Hz.

As shown in Figure 12, Figure 13 and Figure 14, all cases exhibit cyclic behavior, which is expected given the periodic nature of inhalation and exhalation. The peak velocity magnitudes in Figure 12 are consistent with the values reported in Table 5. Figure 13 demonstrates that both the magnitude and range of pressure increase with higher $V_{th}$, and the corresponding pressure fluctuations become more pronounced. These fluctuations indicate the presence of stronger turbulence and higher vorticity within the cavity, as further illustrated in Figure 14. Increased turbulence and pressure oscillations subject the mucus layer to cyclic shear forces, enhancing shear-thinning behavior and thereby promoting more effective mucus transport. To further elucidate the flow structure inside the nasal cavity, representative streamlines are shown in Figure 15.

This behavior relates directly to the expected effects of RAMPA therapy. By expanding the nasal passages, RAMPA increases airflow rate and air velocity within the cavity. The resulting increases in pressure variation, turbulence intensity, and shear forces all act, individually and synergistically, to enhance paranasal mucus drainage.

#### 3.2.3. Air-Mucus Two-Phase Flow Simulation

In this part, a two-phase flow analysis of air and mucus within the nasal passages is performed. For the purposes of the simulation, the upper portion of the model is assumed to be pre-filled with mucus, as shown in Figure 16a. The discharge is modeled as gravity-driven, and the airflow at the throat is simulated with a steady-state velocity of 1 m/s.

As the results in Figure 16 indicate, the wider nasal passage on the right side of the domain exhibits faster mucus discharge, reflected by the more rapid reduction of the mucus VOF. This observation is consistent with the earlier discussion that shear-thinning behavior contributes to improved mucus drainage. As time progresses in the simulation, the downward movement of mucus accelerates, demonstrating the combined influence of passage width and rheological properties.

Considering the effect of RAMPA therapy, the expansion of the nasal passages produced by RAMPA increases airflow rate and velocity (as discussed in the Introduction). The resulting faster and greater airflow further decreases mucus viscosity through shear-thinning, thereby enhancing drainage efficiency.

From a fluid-mechanics perspective, particularly for highly viscous flows such as mucus (or even pus), increasing the airway cross-section also reduces the effective viscosity near the wall because the boundary-layer effect is diminished. In the context of RAMPA therapy, this indicates that nasal-passage expansion and shear-thinning behavior act synergistically to promote more effective mucus clearance from the sinuses.

### 3.3. Patient Case Report

An 8-year-7-month-old patient with ENT-diagnosed chronic empyema exhibited complete paranasal sinus opacification on CBCT. RAMPA, connected with the gHu-1 intraoral device, was worn for 12 h per day. By age 11 years 7 months, paranasal opacities had resolved, and the ENT physician confirmed improvement in the empyema. Occlusion improved (overjet from 2.7 mm to 1.2 mm; overbite from 1.8 mm to 0.3 mm), and intermolar width (6 + 6) increased from 39.8 mm to 46.5 mm. Figure 17 provides clinical evidence of the patient’s condition before and after RAMPA treatment.

## 4. Limitations of This Research

While several studies have demonstrated that RAMPA therapy can produce promising anterosuperior maxillary expansion, it must be acknowledged that the medical evidence presented here is limited to a single case.

It should also be noted that the geometry used in this study includes minor modifications, such as locally narrowed regions and bends, that were intentionally introduced to better investigate airflow and mucus transport under varying passage conditions. Despite the changes, the model remains representative of a real case and well-suited to the study’s objectives.

## 5. Future Work

Prospective case series incorporating patient-specific CFD models generated from pre- and post-treatment CBCT, together with objective measures of nasal resistance (e.g., active anterior rhinomanometry) and standardized symptom scores, would allow stronger causal inference and provide quantitative validation of the predicted pressure–velocity mechanisms.

## 6. Conclusions

This study demonstrates that the Right Angle Maxillary Protraction Appliance (RAMPA), when combined with semi-rapid maxillary expansion (sRME), improves both the structural and aerodynamic conditions of the upper airway, thereby enhancing paranasal mucus drainage. The integration of finite element mechanics and computational fluid dynamics provided complementary insights: FEM analysis revealed that RAMPA shifts the center of maxillary rotation anterosuperiorly, producing nearly parallel and uniform lateral displacements along the palatal plane. This geometric modification widens the nasal passages while preserving their continuity, avoiding sharp curvatures that can hinder airflow. CFD simulations confirmed that such expansion increases airflow velocity and pressure differentials, generating a suction effect that facilitates mucus clearance from the frontal, ethmoid, maxillary, and sphenoid sinuses. Furthermore, sinusoidal pressure fluctuations during respiration and the shear-thinning behavior of mucus act synergistically to promote drainage and prevent stasis. Clinical validation through a pediatric case demonstrated resolution of paranasal opacities, expansion of occlusal and intermolar widths, and measurable improvement in sinus ventilation. Together, these findings substantiate the hypothesis that RAMPA-assisted sRME provides not only orthopedic correction of maxillary deficiency but also functional enhancement of upper airway dynamics. Although the current evidence is limited to a single case, the consistent agreement between engineering analyses and clinical outcomes supports RAMPA therapy as a promising adjunctive treatment for pediatric patients with sinus-related respiratory impairment. Future studies with larger cohorts and quantitative nasal resistance data are warranted to consolidate these results.

## Figures and Tables

**Figure 1 biomimetics-11-00005-f001:**
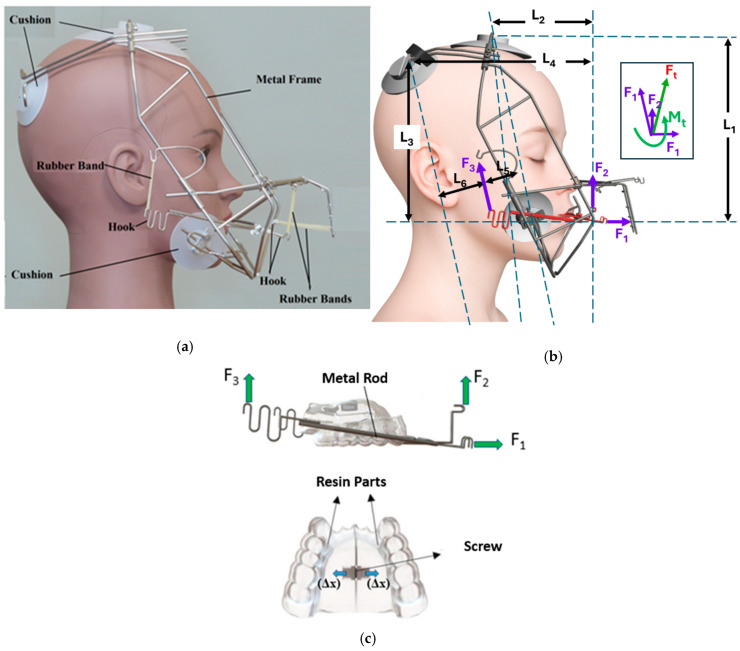
(**a**) Schematic of RAMPA worn by a manikin; (**b**) diagram of external forces applied on RAMPA; (**c**) gHu-1 intraoral appliance from the top view.

**Figure 2 biomimetics-11-00005-f002:**
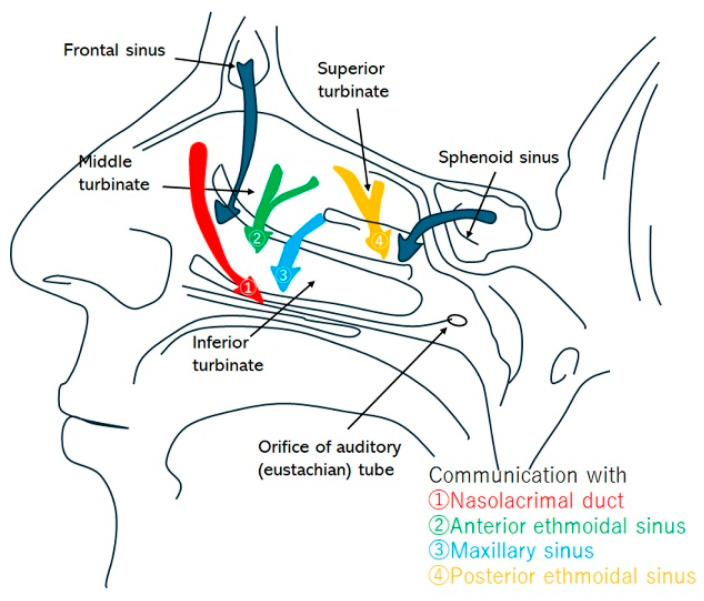
Schematic routes of mucus drainage from the sinuses.

**Figure 3 biomimetics-11-00005-f003:**
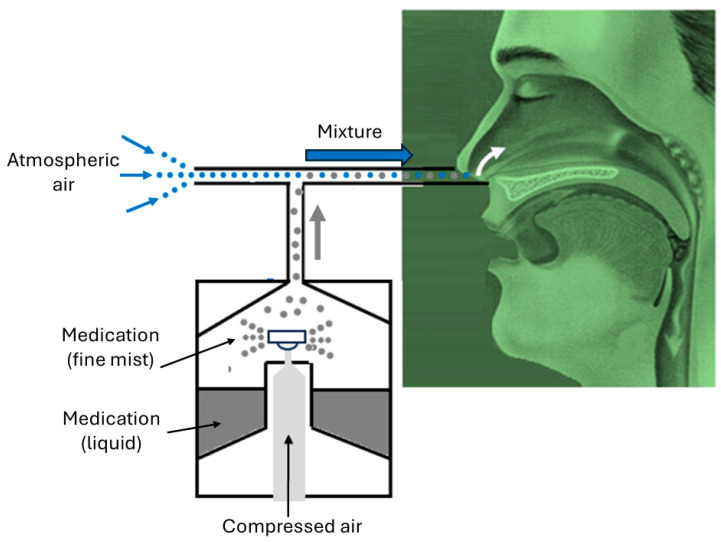
Application of Bernoulli’s principle to a nebulizer.

**Figure 4 biomimetics-11-00005-f004:**
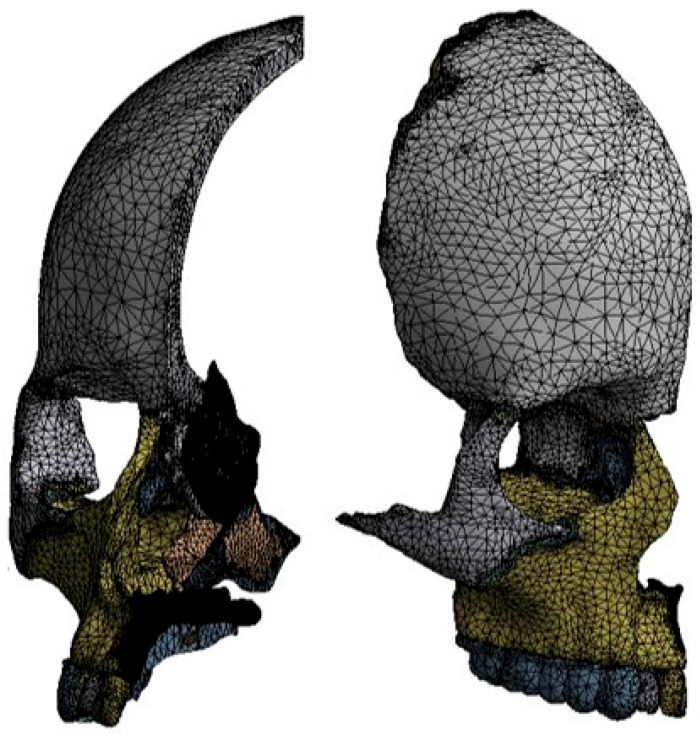
Mesh arrangement employed for the FEM simulation.

**Figure 5 biomimetics-11-00005-f005:**
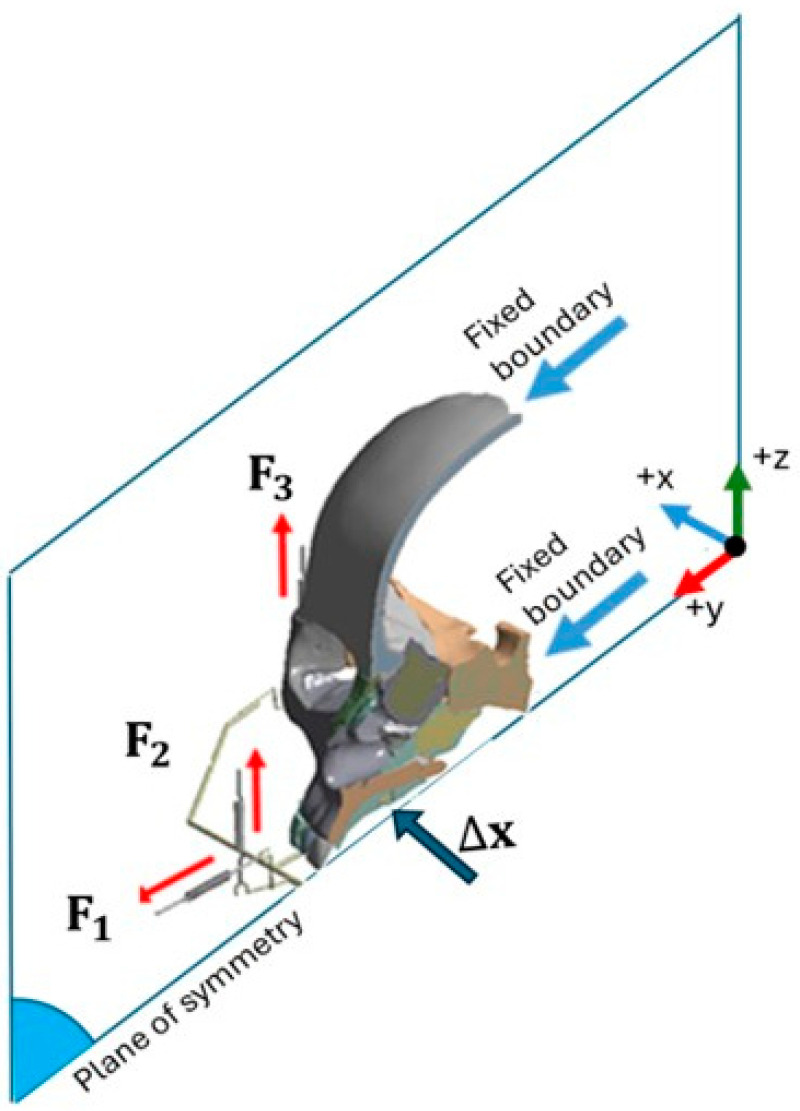
Boundary conditions and forces applied to the FEM simulation.

**Figure 6 biomimetics-11-00005-f006:**
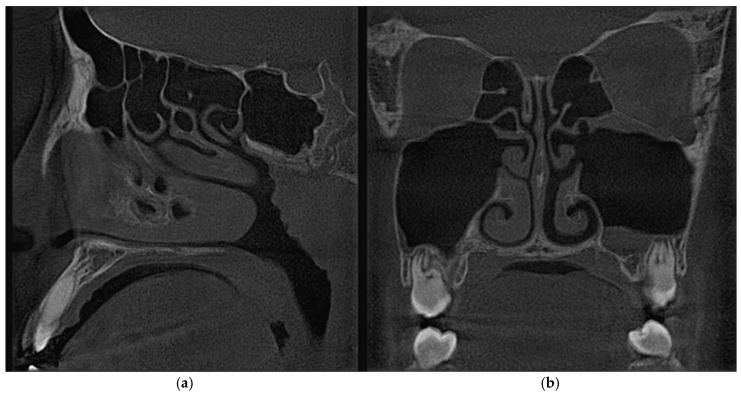
CT scan images of a patient. (**a**) The sagittal view. (**b**) The frontal view.

**Figure 7 biomimetics-11-00005-f007:**
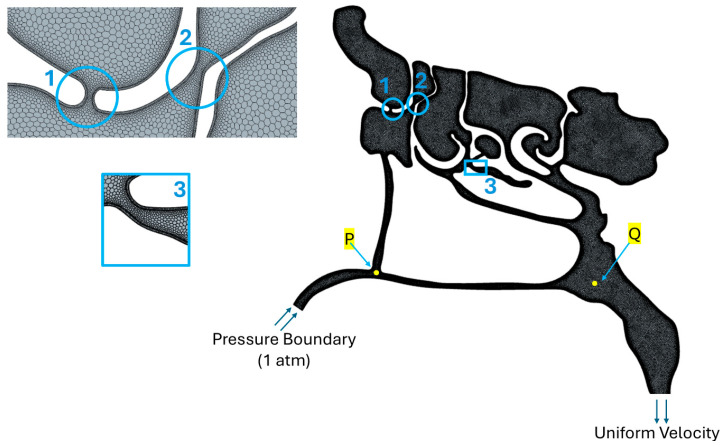
Mesh arrangement used for the CFD simulation includes narrow passages (No. 1 and 2) and a local bend (No. 3). Points P and Q are used for the evaluation of flow filed inside the nasal cavity.

**Figure 8 biomimetics-11-00005-f008:**
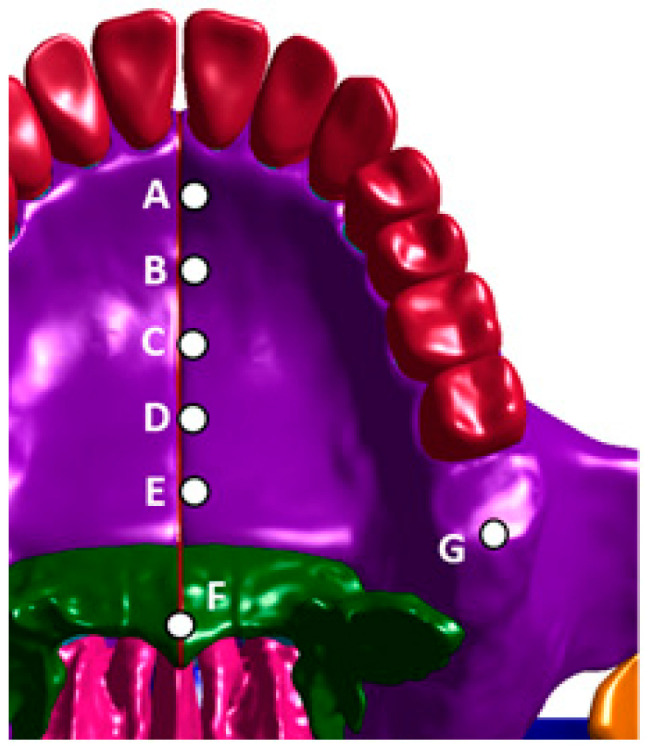
Checkpoints on the MPS: A, point near the incisive foramen; E, point near the palatine bone; B–D points divide the A–E line into 4 quarters.

**Figure 9 biomimetics-11-00005-f009:**
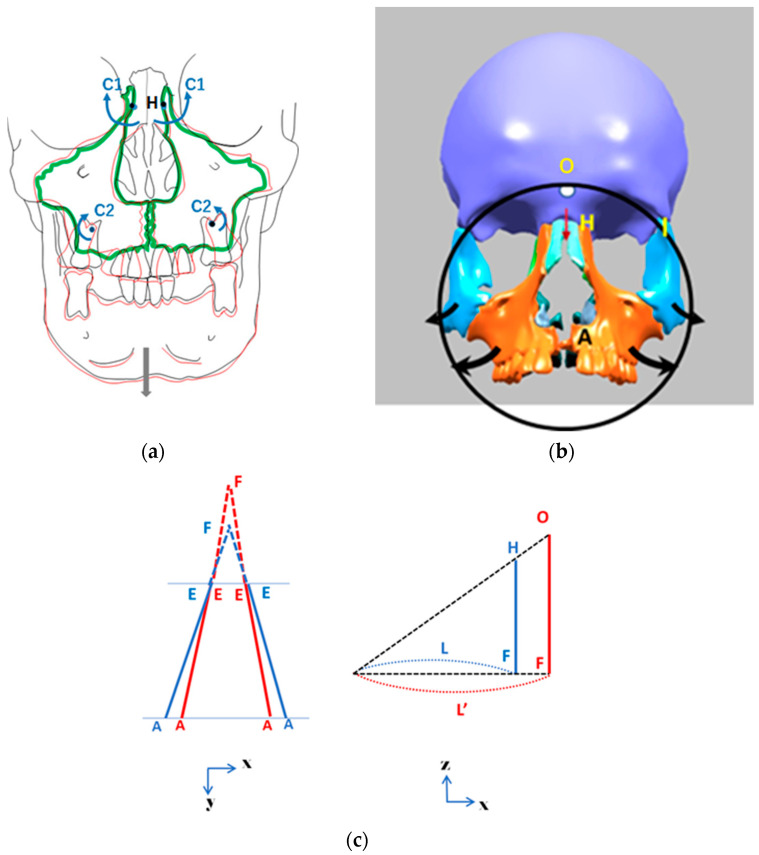
Graphical demonstration of midpalatal expansion as a result of RAMPA therapy with gHu-1 (center of rotation of the maxilla is the frontomaxillary suture (H)). (**a**) With the midpalatal expansion, the center of rotation of the maxilla is the frontomaxillary suture (H). (**b**) Higher center of rotation (O). (**c**) The relationship between centers of rotation, O and H.

**Figure 10 biomimetics-11-00005-f010:**
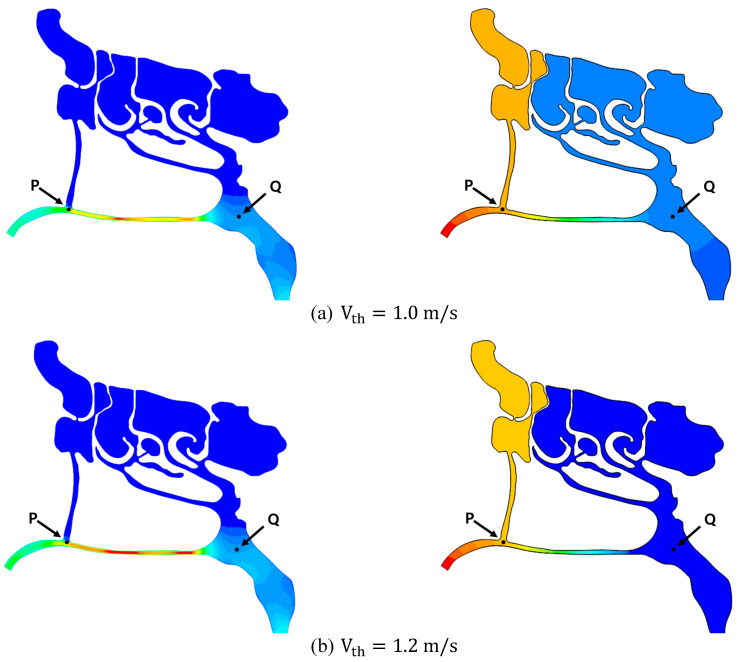
Velocity and pressure distribution of the nasal passage for different throat air velocities. (Uniform colormap scaling (fixed min–max limits) is applied to all plots to permit direct comparison of velocity and pressure distributions between the various simulation cases.).

**Figure 11 biomimetics-11-00005-f011:**
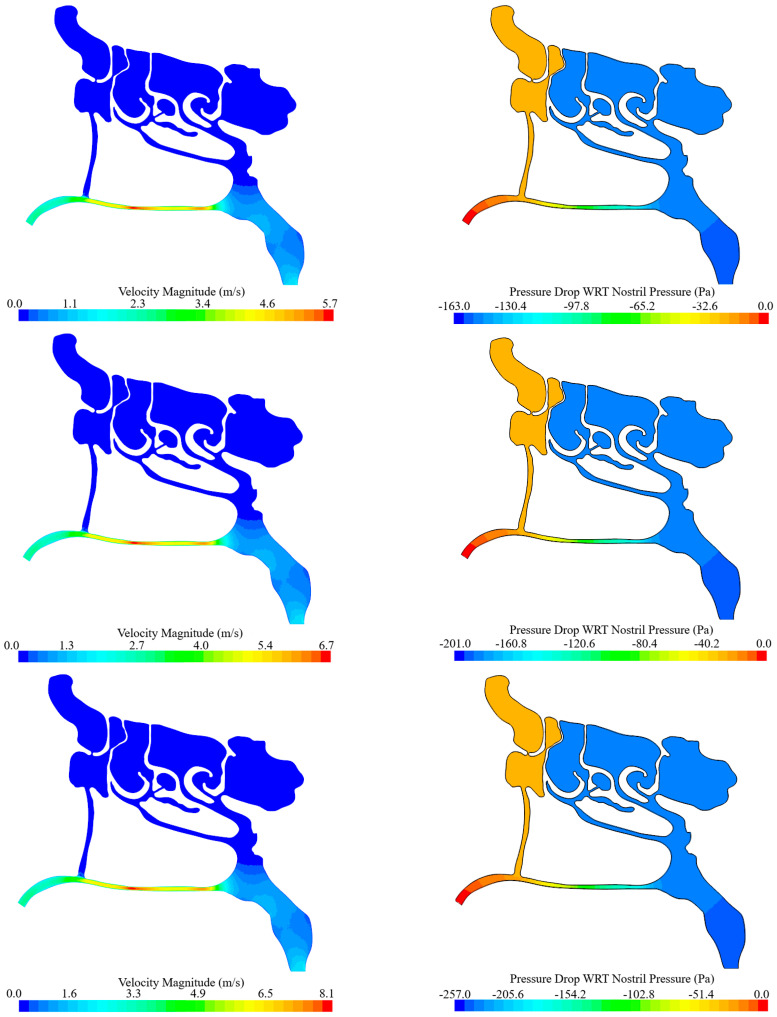
Velocity and pressure distribution of the nasal passage for different throat air velocities (Case-specific colormap scaling, using the local minimum and maximum values for each simulation, is applied to allow clear comparison of the range of velocity and pressure values across different cases.).

**Figure 12 biomimetics-11-00005-f012:**
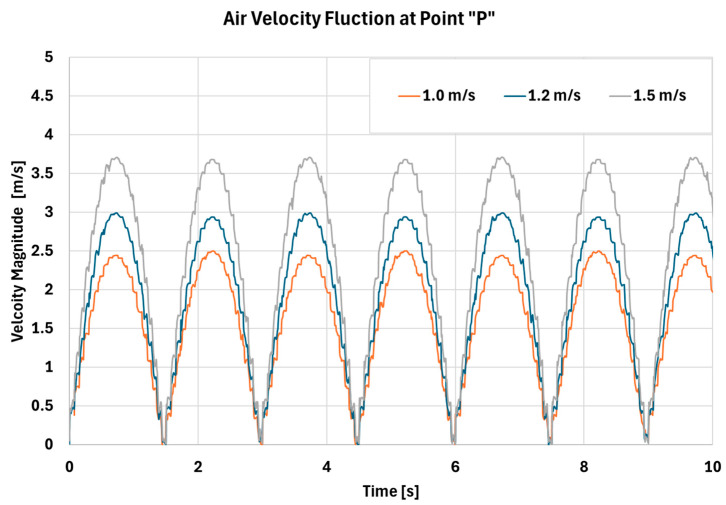
Comparison between velocity magnitude fluctuation at point “P” for different air speeds at the throat location.

**Figure 13 biomimetics-11-00005-f013:**
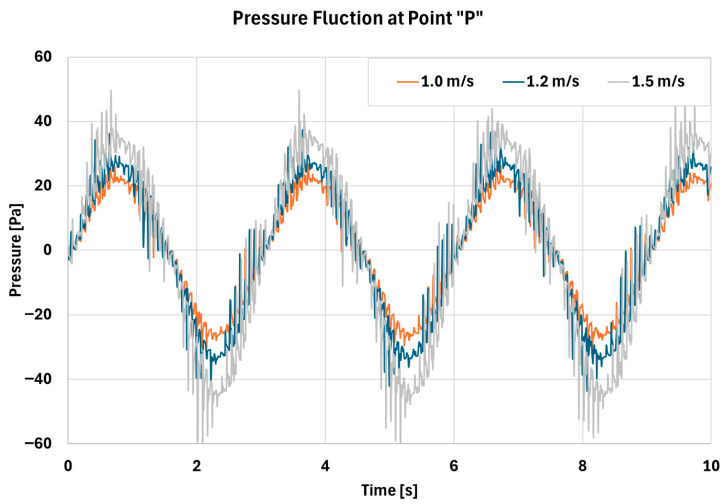
Comparison between pressure fluctuation at point “P” for different air speeds at the throat location.

**Figure 14 biomimetics-11-00005-f014:**
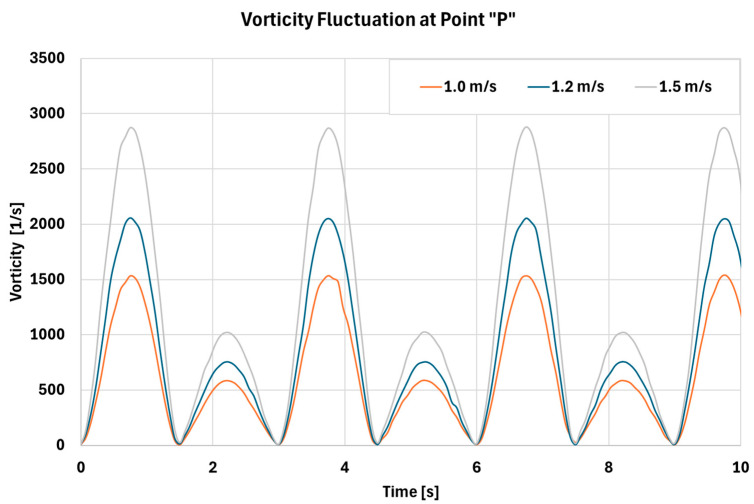
Comparison between vorticity magnitude fluctuation at point “P” for different air speeds at the throat location.

**Figure 15 biomimetics-11-00005-f015:**
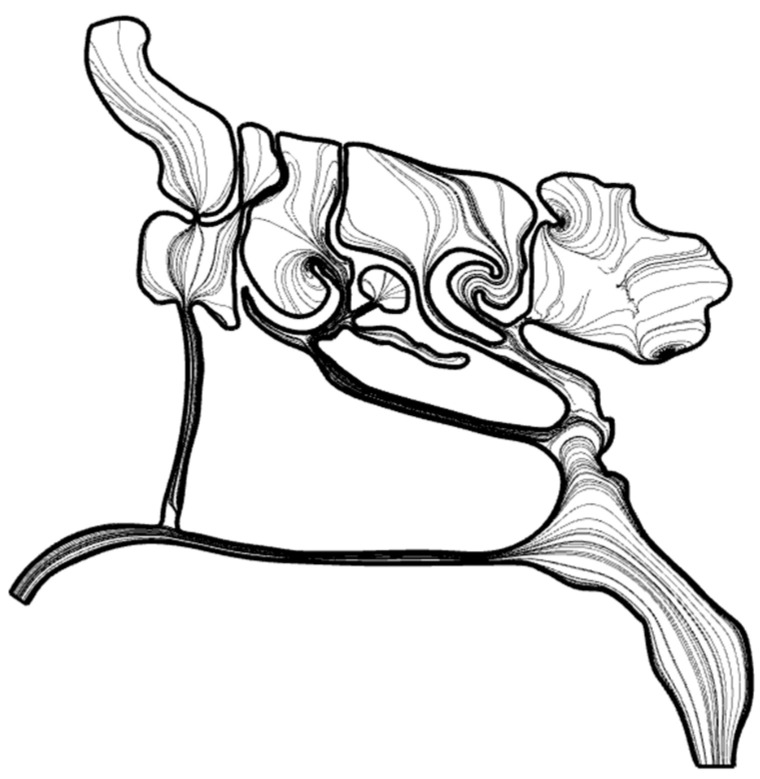
A generic view of streamlines and vortical structures inside the nasal cavity.

**Figure 16 biomimetics-11-00005-f016:**
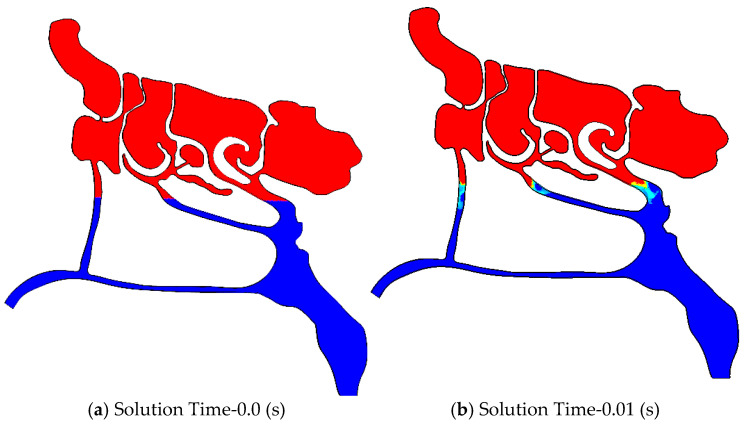
Two-phase flow analysis of air-mucus flow inside the nasal passages at different solution times.

**Figure 17 biomimetics-11-00005-f017:**
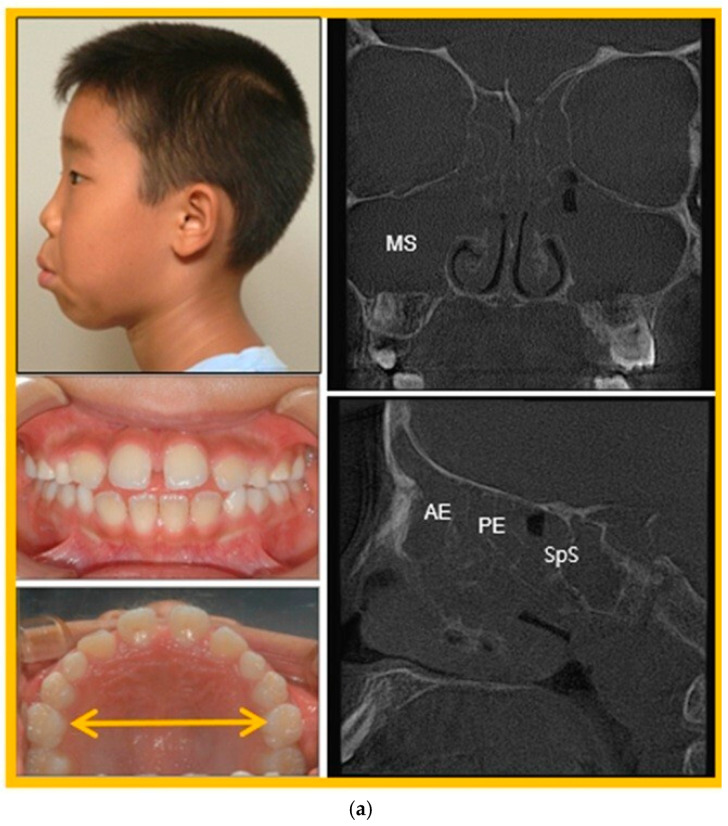
Photographs and CT image between pre- and post- treatment; (**a**) Pretreatment (8y7m), Over Jet 2.7 mm, Over Bite 1.8 mm, 6 + 6 39.8 mm; (**b**) Post-treatment (11y7m), Over Jet 1.2 mm, Over Bite 0.3 mm, 6 + 6 46.5 mm (Abbreviations: MS = Maxillary Sinus, AE = Anterior Ethmoid Sinus, PE = Posterior Ethmoid Sinus, SpS = Sphenoid Sinus).

**Table 1 biomimetics-11-00005-t001:** Canonical routes of mucus drainage.

Sinus	Drainage Pathway	Approximate Size/Features
Frontal sinus	Frontal drainage pathway → direct to middle meatus, or via infundibulum to middle meatus → nasal cavity → nasopharynx	~2–4 mm
Ethmoid bulla	Typically drains posteriorly through retrobullar cleft → middle meatus → nasal cavity → nasopharynx	Variable pneumatization (up to several mm); one of the largest ethmoid air cells
Maxillary sinus	Infundibulum → middle meatus → nasal cavity → nasopharynx	~2–4 mm
Posterior ethmoid air cells	Superior meatus (and supreme when present) → sphenoethmoid recess → nasal cavity → nasopharynx	Several small ostia (<2 mm each)
Sphenoid sinus	Sphenoethmoid recess → nasal cavity → nasopharynx	~2–5 mm

**Table 2 biomimetics-11-00005-t002:** Material properties used for present FEA.

Item	Young’s Modulus (MPa)	Poisson’s Ratio (-)
Cortical bone	13,800	0.26
Cancellous bone	1370	0.3
Periodontal ligament	50	0.49
Teeth	18,600	0.31
Suture (Cartilage)	10	0.49
Acrylic Resin (Orthocryl^®^ DENTAURUM, Ispringen, Germany)	3543	0.3
Stainless steel (AISI 316)	193,000	0.31

**Table 3 biomimetics-11-00005-t003:** Forced and dimensions of RAMPA used for the present case study.

			Dimensions (mm)					
$\left|{\vec{F}}_{1}\right|$	$\left|{\vec{F}}_{2}\right|$	$\left|{\vec{F}}_{3}\right|$	$L_{1}$	$L_{2}$	$L_{3}$	$L_{4}$	$L_{5}$	$L_{6}$
2.94	1.44	4.0	190	105	160	185	35	50

**Table 4 biomimetics-11-00005-t004:** Displacement of points A-E on the mid-palatal line in the g-Hu1 simulation.

Point	With RAMPA (mm)
A (near incisive foramen)	0.151
B	0.144
C	0.138
D	0.128
E (near palatine bone)	0.101

**Table 5 biomimetics-11-00005-t005:** Pressure drop and velocity values at sample points inside the nasal cavity.

Air Speed (m/s)					Pressure Drop WRT to Nostril (Pa)	
	Throat	Nostril	Point “P”	Point “Q”	Point “P”	Point “Q”
Case 1	1.0	1.74	2.44	0.54	−26.11	−149.73
Case 2	1.2	2.11	2.93	0.64	−33.02	−185.42
Case 3	1.5	2.62	3.67	0.78	−44.21	−237.17

## Data Availability

All data supporting the findings of this study are contained within the article.

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
