# Peer review of "Impact of RAMPA Therapy on Nasal Cavity Expansion and Paranasal Drainage: Fluid Mechanics Analysis, CAE Simulation, and a Case Study"

_biomimetics, 2025, doi:10.3390/biomimetics11010005_

Round 1
Reviewer 1 Report
Comments and Suggestions for Authors
The manuscript investigates the aerodynamic and structural effects of RAMPA therapy on sinonasal airflow and mucus drainage using a combination of fluid mechanics theory, CFD simulations (single-phase and two-phase VOF), prior FEM results, and a single patient case. The topic is relevant to craniofacial biomechanics and upper-airway function, and the integration of CFD, FEM, and clinical observations has potential. However, the manuscript in its current form requires major revision. The primary issues are organizational and structural. Please note my following comments:
- Section 2 (“Materials and Methods”) lists four high-level components but does not provide the level of detail expected in this section, rendering the section incomplete. Elements of Materials and Methods are scattered in the Results section, some of which are incomplete and lacks essential details needed for reproducibility (e.g., mesh characteristics, solver settings, mucus properties used in VOF, boundary condition justification, FEM model parameters, CBCT parameters, etc.).
- The Results and Discussion section mixes theoretical background, methodological explanation, and literature review elements with actual results; clearer separation and restructuring are needed.
- Several pages in the Results and Discussion section are devoted to Bernoulli, Poiseuille, Darcy–Weisbach equations, sinusoidal airflow, and mucus rheology. These background equations do not belong in the Results; at most, only a brief conceptual mention should appear in the Introduction. Their current placement disrupts the flow of the actual findings.
- Subsection numbering is inconsistent throughout the manuscript (e.g., repeated “1.1.1” within Section 3), making the document difficult to navigate.
- The airway geometry used for CFD was intentionally modified by the authors to create three artificial passage diameters (“narrow,” “medium,” and “wide,” with an added choke point), as stated in the manuscript. Because this altered geometry does not represent patient anatomy or documented RAMPA-induced anatomical changes, this modification requires stronger justification and a clearer explanation of how (or whether) these artificial diameter variations relate to clinically observed effects of RAMPA.
- Points P and Q should be labeled directly on all contour plots where their values are reported (Figures 8-9). This would help readers immediately identify their locations without referring back to Figure 7.
- Several sentences contain grammatical errors, typos, or unclear phrasing; extensive language revision is required for clarity and readability.
Reviewer 2 Report
Comments and Suggestions for Authors
This research paper investigates the biomechanical effects of the Right Angle Maxillary Protraction Appliance (RAMPA) combined with semi-rapid maxillary expansion (sRME) on nasal airflow and paranasal sinus drainage. The study employs a multidisciplinary approach integrating computational fluid dynamics (CFD), finite element method (FEM) analysis, and clinical case validation to demonstrate how RAMPA therapy enhances mucus clearance from paranasal sinuses. Through CFD simulations of single-phase (air) and two-phase (air-mucus) flows, the researchers show that nasal cavity expansion increases airflow velocity and creates pressure differentials that generate suction effects promoting frontal sinus mucus clearance. FEM analysis reveals that RAMPA shifts the center of maxillary rotation anterosuperiorly, producing parallel-to-palatal expansion that widens nasal passages while maintaining smooth geometry. The clinical validation involves an 8-year-old patient with chronic empyema who showed complete resolution of paranasal opacities and improved occlusal measurements after RAMPA treatment. The study concludes that RAMPA-assisted sRME provides both orthopedic correction and functional enhancement of upper airway dynamics through biomimetic optimization of natural sinonasal drainage mechanisms.
The paper demonstrates strong technical content but suffers from several writing and presentation issues that detract from its overall quality. The writing contains numerous grammatical errors, awkward phrasing, and inconsistent terminology throughout (e.g., "RAPMA" vs "RAMPA," incomplete sentences, and unclear pronoun references). The structure is somewhat disorganized, with methods and results sections that could benefit from clearer subsection headings and more logical flow. While the integration of CFD and FEM analyses with clinical validation is methodologically sound, the presentation of results lacks clarity in some figures and tables, and the statistical analysis is limited given the single case study design. The biomimetics framework, while conceptually interesting, feels somewhat forced and could be better integrated into the narrative. Additionally, the paper would benefit from more rigorous peer review to address the numerous typographical errors and improve the overall readability for an international scientific audience.
Please see my detailed comments below.
- It might be better to describe the model development and validation in Materials and Methods section instead of having that information spread through the Results and Discussions section.
- Please define abbreviations when first use them, e.g., RME and FEM
- The numbering of each subsection is inconsistent in the Results and Discussions section. Please correct.
- For the statement “… the model is slightly changed in such a way that the air passages mimic three different diameters (narrow diameter with a chock point in the middle, medium diameter in left and wider diameter in the left”, what is chock point and does Figure 6 show chock point in the middle? What does medium or wider diameter in left mean? Are then medium and wider diameter cases shown in Figures? Please clarify.
- It could benefit readers with CFD background by providing more details of CFD meshing configuration, such as mesh size, quality, number of prism layers and mesh independence study.
- What is the air volume flowrate when 𝑉𝑡ℎ = 1 𝑚/𝑠?
- If I understand it correctly, in section “Effect of sinusoidal respiration on pressure fluctuation inside nasal cavity”, it simulated both inhalation and exhalation processes. If that is the case, then in Figure 10, it suggests replacing y-axis title “Velocity” with “Velocity Magnitude” as only positive values are shown in the graph. Also, can the authors confirm that the x-axis ranges from 1 to 800 seconds? What is the time period of one breathing cycle?
- Why in Figure 10, the 3 cases have a phase shift about 1/3 second?
- In section “Two phase flow simulation (air-mucus) using VOF method”, does “… the air speed inside the nasal passage is steady state of 1m/s” mean 1m/s at the throat or at which location?
- In Figure 14, are those results from multiple simulated cases with different diameters? To me, it shows one simulation case at various time stamps. Why do the authors claim that “… in Figure 14, a wider passage diameter results in a faster mucus discharge”?
Round 2
Reviewer 1 Report
Comments and Suggestions for Authors
The manuscript is substantially improved and my previous concerns have been addressed. I would like to only note one minor typographical correction: in the caption of Figure 7, “CFD imulation” should be updated to “CFD simulation.”
Reviewer 2 Report
Comments and Suggestions for Authors
The revised manuscript has resovled all my previous comments and concerns related to the original manuscript. The publication of this paper in Biomimetics is recommended.